# Analysis of factors impacting electoral integrity in Africa between 2006–2023 - Examining the association between free and fair election and rule of law

**Michael Oluchi Ikoko**[ID]°*, **Dilek Latif**°

Department of International Relations, Cyprus International University, Nicosia, Cyprus

° These authors contributed equally to this work.
* michaelikokuo@gmail.com

## Abstract

Regular elections in Africa are associated with increasing democratization. However, the level of electoral integrity has sparked post-electoral violence in several African countries. Furthermore, the rise of military regimes in West Africa has interrupted the democratic process in some countries. To contribute to understanding democratization, this article explores the impact of the rule of law, electoral pluralism, and political rights on free and fair elections from 2006–2023. The study used the ordinary least squares regression and wavelet quantile on quantile granger causality to examine the predictive power of the rule of law, electoral pluralism, and political rights on free and fair elections to measure electoral integrity. The results show that the three variables predict free and fair elections at different levels. The rule of law was significant at all lower levels of free and fair elections; the rule of law and electoral pluralism were both significant at moderate and higher levels. Hence, African countries with poor electoral process democracies should establish more mechanisms towards the rule of law, and a moderate level of electoral integrity should enhance the established mechanisms.

## 1 Introduction

There is no democracy without regular elections; however, the electoral processes in African countries must be reexamined to ensure effective contributions to democratization. The transformation of self-rule into strong institutions and governance through elections increases democratization. Citizens' participations in elections can be to run for political offices or choose representatives. When elections are conducted with integrity, civic engagement is deepened, the public is educated, competition increases, outcome receive wider legitimacy and government response to the needs of the people is strengthened [1]. However, these expectations are unrealized

**Data availability statement:** All relevant data are submitted along with the manuscript as Supporting information files.

**Funding:** The author(s) received no specific funding for this work.

**Competing interests:** The authors have declared that no competing interests exist.

because of the increasing irregularities during elections. Elections are marred with fraud, manipulation, intimidation, vote buying, and distrust before, during and after the elections. Furthermore, introducing conditions that favor incumbents, creating artificial problems for other candidates, restricting ballot access to supporters of the incumbent, and lacking trustworthy electoral officials jeopardize the purpose of elections. It is, therefore, necessary to avoid the fallacy of electoralism [2].

However, the association of the factors that affect the fairness of elections has remained an ongoing discourse because free and transparent elections are essential to democracy [3]. The first elections in the early years of post-colonial African countries reflected democratic transition with promises of consolidating democracy. The failure to deliver free and fair elections has led to the fallacy of African electoralism expressed through authoritarian regimes' mimicry of democratic procedure by conducting regular elections. Therefore, it is crucial to measure the elections against other democratic conditions within the country because the façade of election might not be associated with the rule of law and political rights. Therefore, understanding the association between the conditions and factors creates a direction for improving democracy [4].

Regular elections in African authoritarian countries have triggered debates over the conditions that contribute to electoral integrity. Elections were held in both the colonial and post-colonial eras in Africa. However, in the colonial era, elections were only for a few legislative seats through which the colonialists would govern the colonies [5]. The elections in post-colonial era featured all electoral offices in the parliamentary and presidential systems. This expansion of democratization created an expectation of continuous inclusion of citizens in self-rule. Apart from most first elections after independence, the results of subsequent elections have been questioned as autocratic regimes emerged in Africa. Rather than incumbents conceding to losing an election, the electoral rules, office tenures, and results were manipulated to retain and consolidate power [6]. Hence, there's need to examine the factors that can improve free and fair elections.

More problematic is the increasing number of failed and successful coups rather than improving electoral integrity. The long-standing autocratic governments were toppled by the military, which complicates the problem and requires a transition to democracy through new elections [7]. While a mix of democratizing countries characterized Africa, it increasingly harbored autocratic civilian and military regimes.

Using levels of electoral fraud, electoral violence, and voter registration integrity, V-Dem data of 2024 presented Nigeria, Ghana, South Africa, Egypt, Kenya, Benin, Botswana, Cape Verde, Cote d'Ivoire, Gambia, Lesotho, Liberia, Madagascar, Malawi, Mauritius, Namibia, Senegal, Tanzania, Uganda and Zambia as top democratizing countries in Africa. The cases of South Africa, Nigeria, Botswana, and Cape Verde shows successive elections and increasing democratization. The Liberian 2023 election highlighted the citizens' will and significant legitimacy given the public opinion that certified the result and welcomed President Joseph Boakai as opposed to former President George Weah. Similarly, the 2024 election in Botswana, which ended the

rule of a party that had governed the country since independence, portrayed increasing democratization in governance, signaling remarkable openness of political space.

In other cases, like Cameroun, elections are significantly staged to mock democratic practices. In Zimbabwe, post-election accusations over electoral fraud have led to an outbreak of conflict in the country, while the post-election period was characterized military takeover in Niger and Burkina Faso [8]. After the National Assembly election, a military junta ousted the Malian president in August 2020 with the promise of a transitional government, which remains unfulfilled [9]. In September of the following year, the Guinean President, Alpha Conde, was deposed by a military leader, Colonel Mamady Doumbouya. A month later, an unconstitutional change of government by General Abdel Fattah al-Burhane led to war in Sudan, which created military factions seeking to gain political power. Burkina Faso's armed forces installed Captain Ibrahim Traore as the transitional president after orchestrating a coup in January 2022 against an incumbent civilian government. Following the same trend, Nigerien General Abdourahamane Tiani deposed the civilian government under President Mohamed Bazoum in July 2024 and established an uncontestable position as the political power of Niger [10]. This study is significance of this discourse because it aims to examines government factors affecting electoral integrity given the pressure to return to civilian rule or improve democratic processes.

Although, there have been pressures against these forms of autocratic regimes to democratize by improving the conduct of elections, however, beyond the pressures, the factors that affect electoral integrity must be understood to contribute to improved democratization in Africa. The multiparty election has been encouraged to improve democratization in Africa to mitigate continuous democratic backsliding integrity [11]. In the early years of independence, a multiparty system was suggested and implemented; however, all African countries' elections were dominated by single or two parties. These promises regarding the contribution of multiparty elections faded in light of the failure to ensure civil and political liberties alongside crucial conditions for democratization, rule of law, and institutional accountability [12]. Therefore, this study does not include the multiparty system as a factor but focuses on the predictive power of the rule of law, electoral pluralism, and political rights on free and fair elections as a measure of electoral integrity.

Electoral integrity is a cornerstone of democratic governance, yet it remains a significant challenge in many African nations. Between 2006 and 2023, Africa experienced a mix of electoral successes and failures, with free and fair elections often overshadowed by allegations of fraud, voter suppression, and weak institutions. Understanding the factors influencing electoral integrity is crucial, particularly in the context of the rule of law. Recent analyses underscored the critical role of robust legal frameworks and independent judicial bodies in ensuring electoral integrity across African nations [12]. Restriction of civic space, such as limiting freedoms of expression, assembly, and association, undermines democratic processes and highlights the necessity of an enabling environment for free and fair elections.

Furthermore, adherence to legal frameworks and the institutionalization of succession have been identified as pivotal factors in maintaining electoral integrity and public trust in governance structures [13]. The rule of law plays a significant role in fostering trust in governance and ensuring political stability, as it safeguards against manipulation of electoral processes and supports fair electoral competition. Given the diversity of electoral experiences across the continent, analyzing the interplay between free elections and the rule of law provides actionable insights for strengthening African democratic systems. Does rule of law, electoral pluralism and political rights improve Electoral Integrity?

The research aim is to shed light on the systemic factors that bolster or undermine electoral integrity by examining the relationship between free and fair elections, and the rule of law, electoral pluralism and political rights.. This study is significant because it addresses a pressing issue for African democracies to ensure credible elections in the face of institutional and political challenges. The findings are expected to guide policymakers, election observers, and civil society organizations in promoting practices that enhance transparency and accountability in the electoral process. Additionally, by focusing on data spanning 17 years, this study provides a comprehensive view of trends and patterns, contributing to a deeper understanding of the evolving dynamics of electoral governance in Africa. Furthermore, the study seeks to inform efforts to safeguard democratic principles and enhance political legitimacy across the continent. Lastly, the study employed Panel

Quantile Regression as it allows exploring how the rule of law influences electoral integrity not just on average but across the full distribution of integrity scores—capturing whether its impact differs in countries with low versus high levels of free and fair elections. By modeling conditional quantiles rather than the mean, this approach accommodates heterogeneity, non-normal errors, and potential outliers common in cross-country panel data, and it reveals whether improvements in the rule of law yield stronger gains in electoral integrity at specific points (e.g., bottom or top deciles) that would be obscured by conventional fixed- or random-effects models.

The structure of the paper is organized as follows: Section 2 presents a comprehensive review of the relevant literature. Section 3 outlines the data and methodology employed in the study. Section 4 discusses the results and key findings. Finally, the last section provides the conclusion, summarizing the study's insights and implications.

## 2 Literature review

Among factors contributing to electoral integrity, the multiparty electoral system has been a common recommendation by scholars and practitioners. As a political framework in which multiple political parties compete for power in elections, it offers voters a range of choices across the ideological spectrum [14]. The reflection of a multiparty system as an essential element of democratic governance fosters political pluralism. Dahl's pluralism explains that multiparty systems contribute to free and fair elections by limiting the dominance of a single party, enhancing transparency, and encouraging accountability [15]. Dahl explored the relationship between multiparty systems and democratic governance as it increases representation. Multiparty systems, therefore, enhance inclusivity by representing a range of interests, thus reducing the likelihood of any group being marginalized in a multiethnic community. This inclusivity supports electoral fairness by ensuring a broad representation of the electorate.

Nevertheless, empirical research on the impact of multiparty systems on electoral fairness shows mixed results. On the one hand, studies like those by Lindberg and Steytler suggest that the introduction of multiparty systems in emerging democracies, especially in Africa, has led to improvements in electoral competition and political participation over time [16,17]. These studies found that the opposition gained political power, strengthened electoral institutions, and improved the quality of elections as multiparty systems became more entrenched. In the case of Nigeria, an incumbent president lost an election to the opposition party after over 16 years of the ruling party's government, reflecting an entrenchment of the system. Nevertherless, government changes do not reflect a free and fair election without considering the driving factors.

However, while multiparty systems theoretically promote fair elections, the existence of multiple parties does not guarantee electoral integrity. Przeworski et al. emphasized that competitive elections require broader conditions, including a free press, an independent judiciary, and the absence of coercion and violence [18]. Even in countries with multiparty systems, elections are still plagued by fraud, voter intimidation, the misuse of state resources for partisan purposes, and compromising their fairness [19]. Sub-Saharan Africa provides several case studies illustrating this paradox. Cameroon, Zimbabwe, and Congo, for instance, have multiparty systems but are frequently criticized for conducting elections that are neither free nor fair [20,21]. Levitsky and Way describe these regimes as "competitive authoritarianism," where the ruling party allows opposition parties to exist but manipulates electoral processes through media control, vote-buying, and gerrymandering. In such contexts, elections may appear competitive, but the playing field is skewed towards the incumbent [22]. Kenya's multiparty system has been marred by ethnic polarization, electoral violence, and accusations of rigging [23]. Similarly, Nigeria's multiparty system has been characterized by electoral fraud, including ballot stuffing and voter intimidation, which have undermined the legitimacy of its elections. These cases underscore that while a multiparty system is necessary for democratic elections, it is insufficient [24]. Free and fair elections depend on other factors, including the integrity of electoral institutions, respect for the rule of law, and democratic political culture.

Discussion on protecting electoral integrity in emerging democracies, focusing on political institutions and processes. Recent studies examined the viability of using photo quick count to improve electoral integrity which argued that using photos and voice

as auditory measures upholds electoral integrity; however, electoral pathologies that undermine integrity persist [20,21]. Additional studies present that there is an increase in the legitimacy of electoral processes in emerging democracies, and fairness must move beyond the ballot. The analysis of electoral integrity perception in 18 countries from 2004 to 2021 was examined [23,24].

Most of these studies examined emerging democracy while this study focused on the most performing democracies in Africa. While other studies examined factors such as auditory measures within countries by focusing on impacting certifying elections by citizens and on electoral integrity perception, this study contributes to reexamining direct factors that impact electoral integrity rather than citizens satisfaction or perception.

### 2.1 Rule of law

The rule of law is crucial in maintaining electoral integrity by ensuring that the legal framework governing elections is impartial, transparent, and consistently applied [25]. In countries where the rule of law is weak, electoral processes are often marred by irregularities, including vote-buying, voter intimidation, and manipulation by powerful elites. According to Norris, the independence of judicial institutions and enforcing electoral laws are critical in safeguarding free and fair elections. When the rule of law is undermined, it erodes trust in the electoral process and leads to post-election violence, as seen in Kenya's 2007 elections. The lack of enforcement mechanisms and accountability for electoral malpractices creates an environment where manipulation is pervasive, further deteriorating the quality of elections [26]. Despite reforms, many developing countries still struggle to establish the legal independence required for fair elections, indicating the need for improving how electoral laws can be better enforced to ensure integrity. Norris explained that the rule of law positively affects electoral integrity by ensuring fair enforcement of electoral laws and safeguarding against manipulation and fraud [27,28]. It indicates that judicial independence strengthens electoral processes by holding political actors accountable and preventing undue influence. Therefore, a well-functioning judicial system, free from interference, guarantees that electoral disputes are handled fairly and transparently, promoting trust in the electoral system [19,27].

### 2.2 Electoral pluralism

Electoral pluralism refers to the extent to which a political system allows multiple parties and candidates to compete freely in elections. Pluralism is considered a fundamental component of electoral integrity, as the absence can lead to a monopolization of political power. As Przeworski et al. suggest, true democracy is one where parties and candidates have an equal chance of losing or winning elections [29]. However, in many electoral systems, pluralism is constrained by incumbents who manipulate the rules of the game to prevent genuine competition through tactics like gerrymandering or restricting media access for opposition candidates. Electoral pluralism also relies on electoral management bodies to create a level playing field [30]. Research has shown that increasing pluralism in African elections has improved their quality. Hence, electoral pluralism positively correlates with electoral integrity by allowing multiple political parties to participate freely in elections. When political competition is robust and different voices can be heard, elections' overall quality and fairness improve. Pluralistic systems are less prone to electoral manipulation as political actors strive for legitimacy through transparent and competitive elections [31].

### 2.3 Political rights

Political rights, such as voting and running for office, are foundational to electoral integrity. When these rights are restricted, either through voter suppression or discriminatory practices, the legitimacy of elections is called into question. Garnett and Zavadaskaya highlight that equal access to participation is a core requirement for a functioning democracy [32]. However, political rights are often curtailed in environments where authoritarian regimes or ruling parties seek to maintain power by limiting voter registration or the candidacy of opposition leaders [6]. The limitation leads to elections that, while procedurally intact, do not reflect the people's will. Political rights violations are positively associated with lower perceptions of electoral integrity, particularly in countries where civil liberties are restricted [33].

To ensure that the rule of law was distinct from potential confounding variables, this study compared subdimensions of each variable as defined by the Varieties of Democracy (V-Dem) dataset. The rule of law subdimensions—judicial independence, compliance with judiciary decisions, and respect for legal procedures—primarily concern institutional judicial processes and enforcement. This creates a clear functional distinction from the subdimensions of political rights, which emphasize citizens' behavioral and participatory rights in the political process, such as freedom of association, freedom of expression, suffrage, and inclusive participation. Similarly, pluralism is characterized by party-level engagement rather than individual political activities. Its subdimensions—participation of multiple political parties, the existence of opposition parties, and electoral competitiveness—address democratic variations from a supply-side perspective, distinct from the institutional, judiciary-centered nature of the rule of law. Lastly, free and fair elections specifically relates to electoral performance, encompassing subdimensions like electoral fraud, electoral violence, and voter registration integrity, thereby differing from the judicial performance captured within the rule of law dimension [27,28].

## 2.4 Gap in literature

In the empirical literature, several studies have posited the rule of law as a foundational factor upon which electoral rule can be established and upheld [25,33]. In contrast, others have explored the impact of Political pluralism on providing a level ground for electoral candidates as a reflection of electoral integrity [30,31]. Mauk and Norris reflect on political rights' positive association with electoral integrity. While considerable research exists on the factors that influence electoral integrity, assessing the impact of these factors together as independent variables – rule of law, electoral pluralism, and political rights – in the long-run electoral integrity has not been examined in previous studies. Hence, a significant gap remains in under standing the mechanisms that improve electoral integrity in the long run. Therefore, this study contributes to the democratization by examining factors associated with free and fair elections in most democratic African countries [30,31].

## 3 Research methodology

### 3.1 Case selection

Since all African countries run the multiparty system and attempt to focus on electoral plurality and political rights, this study used a panel data set to examine twenty top democratizing countries in Africa – Nigeria, Ghana, South Africa, Egypt, Kenya, Benin, Botswana, Cape Verde, Cote d'Ivoire, Gambia, Lesotho, Liberia, Madagascar, Malawi, Mauritius, Namibia, Senegal, Tanzania, Uganda and Zambia. According to Gerring, using non-random selection procedures that focus on the most similar cases provides deeper insight to a phenomenon [34]. In this article, these are Africa's most democratizing countries, providing valuable insight into the factors that promote electoral integrity. The top 20 democratizers represent a subset of African countries that have held regular elections and achieved a certain level of integrity that contributes to democratic improvement. Therefore, their selection is justified by their performance. These 20 countries constitute approximately 40 percent of all African nations, offering a meaningful and representative sample. This study adheres to all ethical practices throughout the writing process. Approval by the Ethical Committee of Cyprus International University

### 3.2 Variable and data source

As a result of limited research on the impact of the rule of law, electoral pluralism, and political rights on electoral integrity, this current study tends to progress the research by examining the association between Free and Fair Elections and the Rule of Law, Electoral Pluralism and Political Rights. In addition, to resolve the limitation of previous studies that have omitted longitudinal impact on independent variables on free and fair elections, Autoregressive Distributed Lag (ARDL) that provides analysis on short-run and long-run relationships among variables regardless of stationary issues has been employed to examine the long-run effect. Times series data stretching for 17 years (2006–2023) have been collected from V-Dem (2024) for the rule of law, free and fair election, and the Economist Intelligence Unit for electoral pluralism. This study stands on the work of Edgell et al. [35] investigating the global test of the democratization by-election hypothesis,

The econometric model of the above function is written as.

$$FFE_{it} = \alpha + \beta_1 ROL_{it} + \beta_2 EP_{it} + \beta_3 PR_{it} + \in_{it} \tag{1}$$

The data was collected from reliable sources: V-dem database for the Free and fair election, rule of law, Economist Intelligence Unit database for Electoral pluralism, and Freedom House database for Political Rights between the top twenty African countries for 17 years between 2006–2023. Free and fair election, which reflects electoral integrity, is employed as the dependent variable; the rule of law (ROL), electoral pluralism (EP), and political rights (PR) are independent variables.

The panel data regression models have been employed to understand how the variables interact. Fixed and random effects models have been used to explain the time-variants and variables that vary across different time-variants, impacting the relationship between Free and Fair Elections, rule of law, electoral pluralism, and political rights. Several robustness checks have been conducted on independent variables ROL, EP, and PR using econometric methods such as the OLS and dynamics panel data models.

This study provides empirical evidence of the association between free and fair elections and the rule of law alongside the impact of electoral pluralism and political rights in the top 20 democratic countries in Africa.

### 3.3 Econometric model

3.3.1 *Cross-sectional dependency test (CSDT/CDSA).* The cross-sectional dependency test for examining cross-sectional units interdependency has been employed for the detection of cross-sectional dependence and is analyzed through Pesaran's CD technique framework, which is mainly dependent on the cross-sectional residuals augmented test, which is written as:

$$\Delta \widetilde{y}_{it} = \rho \widetilde{y}_{it-1} + \sum_{j=1}^{N} \gamma_j \Delta \widetilde{y}_{jt-1} + \sum_{j=1}^{N} \beta_j \widetilde{X}_{ijt-1} + \in_{it} \tag{2}$$

the Δ stands as the first operator, $\widetilde{y}it$ represents the transformed dependent variable, $X_{ijt}-1$ the independent variables used as conditions $p$ as autoregression, $\gamma_j$ as the coefficient of cross-sectional lag, $\beta_j$ as coefficients for the independent variables, $\epsilon_it$= error term. The cross-sectional units are represented with $N, i$ unit dimension, and $t$ time dimension.

CSDA regression is presented as follows;

$$CD = T \frac{\sum_{i=1}^{N} \sum_{t=1}^{T} \hat{\in} it^2}{\hat{\sigma}^2 \sum i = 1^N \sum_{t=1}^{T} \hat{v}_{it}^2} \tag{3}$$

$T$ denotes the number of periods, $\hat{\epsilon}it$ denotes CSDA regression residual, $\hat{\sigma}2$ refers to the average squared error obtained from cross-sectional dependency regression, and $\hat{v}it$ are the residuals from the ordinary least square regression of $\widetilde{y}it$ on the CS average of $\widetilde{y}i$ and $\widetilde{X}ij$.

The null hypothesis that cross-sectional dependency occurs is rejected when the cross-sectional dependency test shows that the CD test presents a normal distribution. This proves that the data contains cross-sectional dependency and that observations are unrelated. Methods considering cross-sectional dependence, including panel data models similar to the variables structure or spatial econometric models, might be required.

Conversely, the null hypothesis reflecting the absence of cross-sectional dependency is rejected, considering that the estimated CD test statistic is ≤ the crucial value, allowing us to conclude that observations are unrelated.

The test statistic is expected to present a normal distribution; therefore, there is no cross-sectional dependency, and the null hypothesis will be rejected. The null hypothesis that no cross-sectional data exists if the calculated test statistic exceeds the crucial value is rejected. This indicates that there is proof to imply that The data exhibits cross-sectional

dependency, and the observations are not separate. In this instance, techniques that consider cross-sectional dependence, like the model of the data with a common element or spatial econometric models framework, might be required.

### 3.4 Ordinary Least Square regression with one and two-way fixed effects

To investigate the connection between free and fair elections as the dependent variable and rule of law, electoral pluralism, and political rights, this study used Ordinary Least Squares (OLS). This approach estimates the linear relationship among variables by minimizing the sum of the squared residuals. Additionally, this study employed ordinary least square regression, one-way and two-way fixed effects, to account for time-invariant heterogeneity that is not observed and to cover both fixed effects on time and fixed effects on countries.

The models are written as follows:

$$FFE_{it} = \alpha + \beta 1 ROL_{it} + \beta 2 EP_{it} + \beta 3 PR_{it} + \epsilon_{it} \tag{4}$$

The dependent variable representing nation $i$ in the year $t$ is FFE, the intercept is $\alpha$, the coefficients of the independent variables are $\beta 1$ to $\beta 4$, ROL, EP AND PR (UR), and the error term is $\epsilon_{it}$.

In the formula, this study look at the variables that affect a country's Free and Fair Elections (FFE) in a particular year $t$. The electoral integrity for the specific nation $i$ in year $t$ is denoted by the term $FFE_{it}$.

The base level of a free and fair election (FFE) is represented by the intercept, or $\alpha$, given that all independent variables are at zero.

The coefficient for the rule of law (ROL), denoted by $\beta 1$, indicates the impact of changes in ROL on free and fair elections (FFE).

The coefficient associated with electoral pluralism (EP), $\beta 2$, illustrates how changes in EP affect free and fair elections (FFE).

The coefficient for Political Rights (PR) is represented by the symbol 3, which illustrates how changes in PR affect free and fair elections (FFE).

The error term describes the variability that the independent variables cannot explain at the time of research in Free and Free Election (FFE).

**OLS Regression with One-Way Fixed Effect (One-way FE):**

$$FFE_{it} = \alpha_i + \beta 1 ROL_t + \beta 2 EP_{it} + \beta 3 PR_t + \epsilon_{it} \tag{5}$$

**OLS Regression with Two-Way Fixed Effect (Two-way FE):**

$$FFE_{it} = \alpha_i + \gamma_t + \beta 1 ROL_{it} + \beta 2 EP_{it} + \beta 3 PR_{it} + \epsilon_i \tag{6}$$

The OLS regression model's remaining variables are identical. By considering every aspect, these models allow the study to examine how free and fair elections relate to electoral and legal aspects in African countries while considering unobserved time-invariant variability. This approach includes both country-fixed and time-fixed effects.

### 3.5 Quantile regression method

This study will utilize the simultaneous quantile regression technique to calculate the quantiles of the dependent variables concurrently. This method may be useful at different quantiles for examining the relationships between a set of independent factors and several dependent variables. In this dataset of African countries, this study used simultaneous quantile regression to investigate the impact of electoral plurality (EP), political rights (PR), and the rule of law (ROL) on emissions

from free and fair elections at different quantiles. Precisely, the work estimates the conditional quantiles of FFE using the following model:

$$Qyi, t|xi, t\ (\tau)\ =\ \alpha\tau\ +\ x'i, t\beta\tau\ +\ \epsilon i, t, \tau \tag{7}$$

where the conditional quantile of interest is denoted by ξ, FFE by $yi$, and a vector of independent variables, including ROL, FFE, and EP, by $xi,t$, the quantile-specific intercept and coefficient parameters are $\alpha\tau$ and $\beta\tau$, respectively, while the error term is $\epsilon i,t,\tau$.

To explain the possibility of heterogeneity among these African countries, this study further examined the model using one-way fixed effects as presented below;

$$Qyi, t|xi, t\ (\tau)\ =\ \alpha\tau\ +\ x'i, t\beta\tau\ +\ \epsilon i, t, \tau \tag{8}$$

Where the fixed effect is represented by $\gamma i$, reflects the time-invariant differences in the intercepts among African nations. The following is a representation of the two-way fixed effects model:

$$Qyi, t|xi, t\ (\tau)\ =\ \alpha\tau\ +\ x'i, t\beta\tau\ +\ \epsilon i, t, \tau \tag{9}$$

The time-fixed effect, represented by $\delta t$, reveals the time-invariant changes in the intercepts throughout time.

The model has been estimated in Stata to identify the conditional quantiles of free and fair elections by considering any variance across African nations and historical times.

The fixed effects model can also account for unobserved time-invariant differences between countries and historical periods that might impact the association between the independent and dependent variables.

The study used the panel quantile regression as the primary analysis and the panel OLS is used as a robustness check. Employing both methods complements the analytical strength of each. Quantile regression is more flexible and insightful regarding distributional variations, while panel OLS regression provides a benchmark check for the consistency and reliability of the main findings. This dual approach strengthens the rigor of the research, increasing confidence in the conclusions drawn.

## 4 Result

### 4.1 Descriptive statistics

Table 1 presents summary statistics for four variables: FFE, ROL, EP, and PR. Each variable is analyzed across "overall," "between," and "within" categories. The "overall" statistics represent the combined variation across all observations (N = 360), with FFE, for example, having a mean of 0.532 and a standard deviation of 0.240. The "between" statistics capture variation across different entities (n = 20), showing how these variables vary across subjects rather than over time. For instance, FFE has a between-mean standard deviation of 0.225, suggesting some variation in FFE across the entities studied. The "within" statistics reflect the variation over time within each entity (T = 18), indicating how each variable fluctuates within subjects over the observed periods. For example, FFE, within a standard deviation of 0.096, shows relatively less fluctuation over time than its between-entity variation.

### 4.2 Cointegration

Next, this study examined the cointegration between the dependent and independent variables using Westerlund cointegration. Table 2 shows the outcomes from the Westerlund ECM panel cointegration tests to examine the null hypothesis (H0) of no cointegration among the series. With 20 series and three covariates, the test statistics (Gt, Ga, Pt, Pa) are presented alongside their corresponding Z-values, P-values, and robust P-values. The Gt statistic shows a significant

**Table 1. Descriptive statistics.**

| Variable | Category | Mean | Std. Dev. | Min | Max |
|---|---|---|---|---|---|
| FFE | Overall | 0.532 | 0.240 | 0.000 | 0.896 |
| | Between | 0.225 | 0.126 | – | |
| | Within | 0.096 | 0.241 | – | |
| ROL | Overall | 0.587 | 0.206 | 0.082 | 0.911 |
| | Between | 0.195 | 0.234 | – | |
| | Within | 0.080 | 0.349 | – | |
| EP | Overall | 6.278 | 2.352 | 0.000 | 9.17 |
| | Between | 2.199 | 2.296 | – | |
| | Within | 0.961 | 2.277 | – | |
| PR | Overall | 3.247 | 1.532 | 1.000 | 7 |
| | Between | 1.438 | 1.000 | – | |
| | Within | 0.614 | 2.192 | – | |

**Note:** N = 360, n = 20, T = 18

Z-value of −2.502 and a P-value of 0.006, indicating strong evidence against the null hypothesis, suggesting the presence of cointegration. Similarly, the Ga statistic, with a robust P-value of 0.040, also provides some evidence of cointegration. However, the Pt and Pa statistics yield higher P-values (0.290 and 0.923, respectively), which are not significant, indicating weaker evidence for cointegration with these measures. Overall, while some test statistics support cointegration, the mixed results suggest that further analysis may be beneficial to confirm the relationships between the series.

### 4.3 Panel quantile regression

Progressively, the study employed the panel quantile regression. This technique helps in exploring how the rule of law influences electoral integrity not just on average but across the full distribution of integrity scores—capturing whether its impact differs in countries with low versus high levels of free and fair elections. Table 3 displays the panel quantile regression analysis results regarding the effects of the rule of law (ROL), electoral pluralism (EP), and political rights (PR) on electoral integrity measured by free and fair elections (FFE) across three quantiles: 0.25, 0.50, and 0.75. Quantile regression allows us to understand how these independent variables impact electoral integrity at different levels, providing insights beyond the average effects captured by standard regression models. The results show how these variables influence electoral integrity differently in low, median, and high-integrity contexts.

At the 0.25 quantile, representing contexts with lower free and fair elections, ROL shows a significant positive effect with a coefficient of 0.281 ($p < 0.01$), suggesting that improvements in the rule of law are crucial for increasing free and fair elections in fragile electoral environments. This aligns with studies like Norris and Skaaning, which emphasize the importance of the rule of law in reinforcing democratic processes in weaker states. In contrast, EP is insignificant at this quantile, suggesting that electoral pluralism alone does not substantially impact integrity in low-integrity environments. PR, however, is significantly negative, with a coefficient of −0.110 ($p < 0.01$), indicating that restrictions on political rights harm free and fair elections in these contexts, aligning with Dahl's arguments on the importance of political freedoms [33,36–39].

In the 0.50 quantile, representing the median level of free and fair elections, ROL's effect increases with a coefficient of 0.367 ($p < 0.01$), highlighting an even stronger association between the rule of law and free and fair elections at this level. This result supports research by Diamond, who argues that in transitioning democracies, robust rule of law is fundamental to credible elections. EP also becomes significant here, with a coefficient of 0.011 ($p < 0.05$), indicating that pluralism

**Table 2. Cointegration results.**

| Statistic | Value | Z-value | P-value | Robust P-value |
|---|---|---|---|---|
| Gt | −2.76 | −2.502 | 0.006 | 0.020 |
| Ga | −7.293 | 2.335 | 0.990 | 0.040 |
| Pt | −9.174 | −0.553 | 0.290 | 0.180 |
| Pa | −5.365 | 1.429 | 0.923 | 0.220 |

**Note:** 1%, and 5% are depicted by * and **

**Table 3. Panel quantile regression.**

| Variable | 0.25 quantile | 0.50 quantile | 0.75 quantile |
|---|---|---|---|
| **Rule of Law (ROL)** | | | |
| Coefficient (t-value) | 0.281 (5.98)*** | 0.367 (8.86)*** | 0.438 (9.77)*** |
| **Electoral Pluralism (EP)** | | | |
| Coefficient (t-value) | 0.007 (1.27) | 0.011 (2.10)** | 0.024 (4.29)*** |
| **Political Rights (PR)** | | | |
| Coefficient (t-value) | −0.110 (−11.55)*** | −0.102 (−12.08)*** | −0.061 (−6.67)*** |
| **Constant** | | | |
| Coefficient (t-value) | 0.618 (8.49)*** | 0.571 (8.88)*** | 0.394 (5.66)*** |

**Note: *** p < 0.01, ** p < 0.05, * p < 0.1**

begins to impact electoral integrity at median levels positively. This result aligns with Lijphart's view that pluralism fosters democratic competition, essential for fair elections. PR remains significantly negative, suggesting that even at median levels, limitations on political rights hinder perceptions of electoral fairness, as argued by Levitsky and Way [22].

At the 0.75 quantile, representing high levels of free and fair elections, the impact of ROL strengthens further, with a coefficient of 0.438 (p < 0.01). This finding is consistent with Norris, who argues that high-integrity democracies typically have robust rule-of-law frameworks. EP's coefficient also increases to 0.024 (p < 0.01), showing that pluralism substantially supports electoral fairness in well-functioning electoral systems, aligning with Przeworski, who emphasizes pluralism's role in mature democracies. However, O'Donnell cautions that pluralism without sufficient checks can sometimes lead to fragmentation, potentially challenging long-term electoral integrity [33,40,41].

Political rights (PR) remain negatively associated with free and fair elections across all quantiles, though the magnitude of this effect diminishes at higher levels, with a coefficient of −0.061 at the 0.75 quantiles. Hence, it implies that even in high-integrity environments, restrictions on political rights detract from electoral fairness, albeit to a lesser extent. It aligns with studies that advocate for high-quality democracies with unrestricted political rights. Conversely, Linz and Stepan suggest that minor restrictions might occasionally be tolerated to protect democratic stability in specific contexts [42]. For pictorial information, Fig 1 showcases the panel quantile regression results.

### 4.4 Robustness check

Table 4 presents the results of a robustness check using Panel OLS regression to examine the effects of rule of law (ROL), electoral pluralism (EP), and political rights (PR) on the dependent variable, which has a mean of 0.53 and a standard deviation of 0.24. The ROL variable shows a significant positive effect (coefficient = 0.397, p < 0.01), indicating that improvements in the rule of law are strongly associated with increases in the dependent variable. EP also

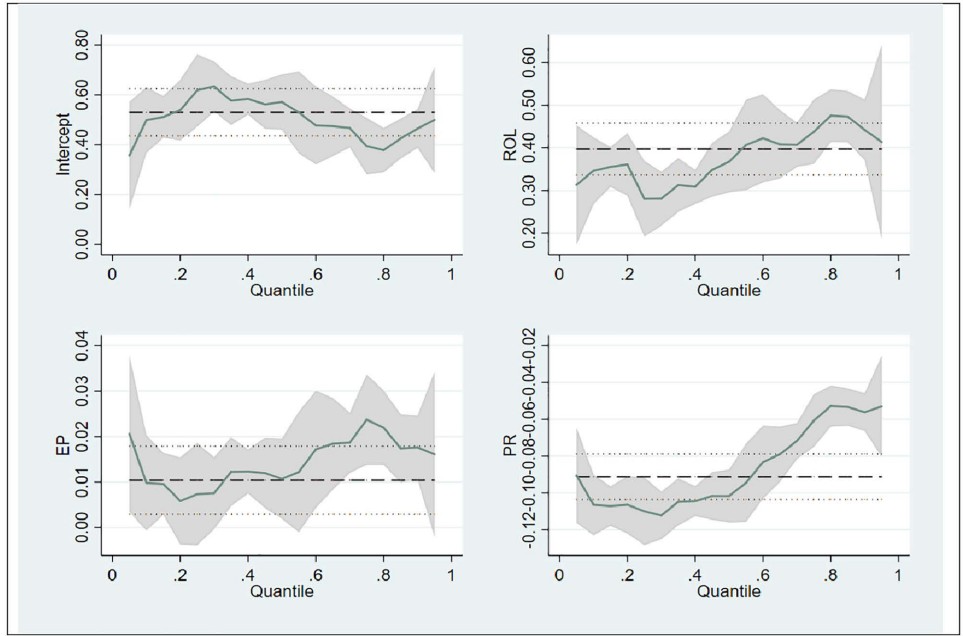

**Fig 1. Panel quantile regression plot.**

has a positive and significant impact (coefficient = 0.01, p < 0.01), though its effect is smaller, suggesting a moderate positive relationship. PR, however, has a significant negative effect (coefficient = −0.091, p < 0.01), implying that restrictions on political rights are linked to decreases in the dependent variable. The model has an R-squared value of 0.84, indicating that these predictors explain 84% of the variance in the dependent variable. The high F-test value (621.35, p < 0.01) confirms the model's overall significance, suggesting that the variables collectively impact the outcome.

### 4.5 Multicollinearity test results

Table 5 displays the Variance Inflation Factor (VIF) results, which assess potential multicollinearity among the independent variables: political rights (PR), electoral pluralism (EP), and rule of law (ROL). PR has the highest VIF value at 3.569, suggesting moderate multicollinearity but staying well below the common threshold of 10, indicating high multicollinearity concerns. EP follows with a VIF of 3.103, and ROL has the lowest VIF at 1.569, indicating minimal multicollinearity. The mean VIF across all variables is 2.747, suggesting that multicollinearity is not likely to be a serious issue in the model, allowing for reliable coefficient estimates.

## 5 Conclusion and policy recommendations

### 5.1 Conclusion

The study highlights the varying impacts of the rule of law, electoral pluralism, and political rights on free and fair elections at different levels among African countries. This suggests that ROL enhances electoral integrity across all African countries' contexts, indicating its foundational role in ensuring fair elections. Without the rule of law, there will be no basis for the electoral process to establish a democracy. Therefore, this study reinforces the role of the rule of law among African countries, as reflected by other scholars [24,27]. Progressively, Electoral pluralism has become more impactful in more stable and developed electoral systems, showing that diversity in political options in the electoral process is especially

**Table 4. Robustness check (panel OLS).**

| Variable | Coefficient (Std. Error) | t-value |
|---|---|---|
| Rule of Law (ROL) | 0.397 (0.031)*** | 12.84 |
| Economic Performance (EP) | 0.010 (0.004)*** | 2.73 |
| Political Rights (PR) | −0.091 (0.006)*** | −14.54 |
| Constant | 0.529 (0.048)*** | 11.04 |

**Note:** *** p<0.01, ** p<0.05, * p<0.1

**Table 5. Variance Inflation Factor (VIF) results.**

| Variable | VIF | 1/VIF |
|---|---|---|
| PR | 3.569 | 0.280 |
| EP | 3.103 | 0.322 |
| ROL | 1.569 | 0.637 |
| Mean VIF | 2.747 | |

valuable when elections are already well-established. However, Political rights, on the other hand, show a negative effect when limited, but this effect is less severe in more stable systems. Therefore, stable African democracies might limit political rights; however, they remain stable due to the rule of law and electoral pluralism, indicating electoral integrity. The study's approach to examining the top 20 democratic African countries stands out because it compares countries within the same continent with similar democratization challenges. Therefore, policy recommendations are well directed to the continent's effort to improve free and fair elections.

## 5.2 Policy recommendations

Enhancing electoral integrity across varying contexts in Africa's top 20 most democratic nations, policy recommendations should target specific challenges identified at different levels of electoral integrity. For African nations with lower electoral integrity (0.25 quantile), strengthening the rule of law (ROL) is fundamental. Investments in judicial independence, transparent law enforcement, and anti-corruption frameworks can help build public trust and ensure fairer elections. Programs to train legal and judicial personnel in upholding democratic values will also reinforce this foundation. Given that electoral pluralism (EP) appears less impactful in these lower integrity contexts, emphasis should initially remain on establishing reliable legal structures to promote electoral fairness [43].

In African countries with moderate electoral integrity (0.50 quantile), the rule of law and electoral pluralism need more attention. Enhancing ROL by bolstering accountability measures and increasing citizen access to legal recourse will contribute to electoral integrity and fair outcomes. Furthermore, as pluralism becomes statistically significant at this level, governments should work to facilitate a multiparty environment by providing equitable media access for opposition parties and ensuring impartial campaign regulations. Such measures foster democratic competition. Additionally, a continuous review of political rights (PR) protections is vital to prevent any restrictions that could damage electoral fairness perceptions [44].

For high-integrity environments (0.75 quantiles), where democracies are more consolidated, policies should strengthen ROL and EP to sustain electoral integrity. Ensuring robust legal protections that prevent abuse of power and enhancing oversight institutions will help safeguard electoral integrity. Since EP has a strong positive impact at this level, further diversifying political representation and promoting inclusive decision-making mechanisms will be beneficial. Policies encouraging constructive dialogue among diverse political groups can foster unity and prevent fragmentation. In these

environments, governments should maintain vigilance over PR to protect electoral integrity, as limitations on political rights, though having a diminished impact, still negatively affect perceptions of fairness [45].

In conclusion, protecting political rights is crucial across all levels of electoral integrity, as restrictions harm free and fair elections at every quantile, albeit to a lesser degree in high-integrity environments. Policies to safeguard freedom of speech, assembly, and political association are essential to support fair electoral systems. Regularly reviewing electoral laws to align with international democratic standards can help protect and expand political rights as these nations' democracies evolve. African countries with low levels of electoral integrity democracies should establish more mechanisms towards the rule of law; others with moderate electoral integrity should enhance the established mechanisms, while nations with high electoral integrity should consolidate and sustain established mechanisms [45].

## Author contributions

**Conceptualization:** Dilek Latif.

**Data curation:** Michael Oluchi Ikoko.

**Methodology:** Michael Oluchi Ikoko.

**Writing – original draft:** Michael Oluchi Ikoko.

**Writing – review & editing:** Michael Oluchi Ikoko, Dilek Latif.

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
