## [Decision Letter · Decision Letter 0]

25 Apr 2025

PONE-D-25-05147Analysis of factors impacting electoral Integrity in Africa between 2006-2023 - Examining the association between free and fair election and rule of law.PLOS ONE

Dear Dr. Ikoko,

Thank you very much for submitting your manuscript to PLOS ONE. After careful consideration, we feel that it has merit but does not fully meet PLOS ONE’s publication criteria as it currently stands. Therefore, we invite you to submit a revised version of the manuscript that addresses the points raised during the review process.

**Following the peer review process, I have carefully considered the reviewers’ comments and, based on their recommendations, I believe that your manuscript requires **major revisions** before it can be considered further.**

**I kindly ask you to review the attached comments from the reviewers and revise your manuscript accordingly. Please prepare a detailed response letter addressing each of the reviewers’ concerns point by point, and clearly indicate the changes you have made in the revised manuscript.**

**We appreciate your contribution and look forward to receiving your revised submission.**

We look forward to receiving your revised manuscript.

Kind regards,

Dr. Mahmut Zeki Akarsu

Academic Editor

PLOS ONE

Journal requirements: 1. When submitting your revision, we need you to address these additional requirements. Please ensure that your manuscript meets PLOS ONE's style requirements, including those for file naming. The PLOS ONE style templates can be found at https://journals.plos.org/plosone/s/file?id=wjVg/PLOSOne_formatting_sample_main_body.pdf and https://journals.plos.org/plosone/s/file?id=ba62/PLOSOne_formatting_sample_title_authors_affiliations.pdf 2. In your Methods section, please include additional information about your dataset and ensure that you have included a statement specifying whether the collection and analysis method complied with the terms and conditions for the source of the data. 3. PLOS requires an ORCID iD for the corresponding author in Editorial Manager on papers submitted after December 6th, 2016. Please ensure that you have an ORCID iD and that it is validated in Editorial Manager. To do this, go to ‘Update my Information’ (in the upper left-hand corner of the main menu), and click on the Fetch/Validate link next to the ORCID field. This will take you to the ORCID site and allow you to create a new iD or authenticate a pre-existing iD in Editorial Manager. 4. Please include captions for your Supporting Information files at the end of your manuscript, and update any in-text citations to match accordingly. Please see our Supporting Information guidelines for more information: http://journals.plos.org/plosone/s/supporting-information.

Reviewers' comments:

Reviewer's Responses to Questions

**Comments to the Author**

1. Is the manuscript technically sound, and do the data support the conclusions?

Reviewer #1: Yes

Reviewer #2: No

Reviewer #3: Yes

2. Has the statistical analysis been performed appropriately and rigorously? 

Reviewer #1: Yes

Reviewer #2: No

Reviewer #3: Yes

3. Have the authors made all data underlying the findings in their manuscript fully available?

Reviewer #1: Yes

Reviewer #2: No

Reviewer #3: Yes

4. Is the manuscript presented in an intelligible fashion and written in standard English?

Reviewer #1: Yes

Reviewer #2: Yes

Reviewer #3: No

5. Review Comments to the Author

**Reviewer #1: ** This is a solid article that advances understanding of why elections in Africa have not produced more robust democracies. The longitudinal approach is welcome. Some improvements could be made. First, there should be a more rigorous discussion of how each variable is measured using V-Dem data to ensure that the rule of law variable is distinct from other confounders. Second, the author(s) should justify their case selection. Why do they limit analysis to the top 20 democratizers and ho might this affect their findings?

**Reviewer #2: ** The abstract: The conclusion of the abstract is too theoretical. We can do without that.

Line 98. We should have a footnote to indicate the figures that support this sentence.

What is the contribution of this article to academic literature?

Clearly, the author doesn't have a single way to cite references in the body of the document.

In the countries that make up your sample, we don't find the countries cited in the introduction. Why?

Abbreviations that are not explained previously. Example:

CSDA, OLS,

The results are not convincing. First, I don't understand the choice of Panel Quantile Regression. The author should provide reasons. Second, he talks about a dynamic panel. We don't see his results in the document, even though the problem of endogeneity (reverse causality) is obvious. Third, we also don't address the endogeneity problem related to omission bias. Fourth, we have tables that look like software outputs. This is not elaborate.

**Reviewer #3:**  Thank you for the opportunity to review this manuscript. The paper addresses an important and timely topic by exploring potential factors that contribute to free and fair elections—specifically, rule of law, electoral pluralism, and political rights. Against the backdrop of limited democratization in many African countries despite the regular holding of elections, the authors seek to better understand which factors help foster elections that genuinely support democratic development.

The research question is highly relevant, and the dataset is well-suited to address it. However, I believe the methodological presentation and interpretation of results would benefit from restructuring to improve overall clarity. I recommend that the authors be given the opportunity to revise and resubmit the paper.

Currently, the empirical section is difficult to follow, particularly for readers who may not be familiar with advanced econometric techniques such as panel quantile regression. Although the models are introduced sequentially, it is not always clear how they relate to one another or why specific methodological choices—such as the use of quantile regression over other approaches—were made. I was also unsure why panel OLS regression appears as a robustness check when earlier models using OLS with one- and two-way fixed effects are presented as main models. Clarifying which models are considered the primary analyses and which serve as robustness checks, and explaining the rationale behind these choices, would certiainly help strengthen the empirical contribution.

Additionally, more detail is needed regarding the group referred to as the "twenty top democratizing countries" in Africa. How were these countries selected? Which data source (e.g., V-Dem?) was used to define this group? Why was the sample restricted to 20 countries? Providing this information would help readers better understand the case selection and its relevance to the broader argument.

Minor comments:

• The manuscript would benefit from careful proofreading and editing for grammar, clarity, and consistency of voice (e.g., switching between “we” and “I”).

• There is some repetition in the discussion of data sources, particularly across Sections 3.1 and 3.2, which can make the argument harder to follow.

• Ensure that all cited works appear in the reference list—for example, Edgeli (2018) is mentioned in the text but not included in the references.

6. PLOS authors have the option to publish the peer review history of their article (what does this mean? ). If published, this will include your full peer review and any attached files.

**Do you want your identity to be public for this peer review?** For information about this choice, including consent withdrawal, please see our Privacy Policy .

Reviewer #1: No

Reviewer #2: No

Reviewer #3: No

---

## [Author Response · Author response to Decision Letter 1]

3 Jul 2025

Reviewers’ Comment Authors’ Responses

Reviewer #1:

1. First, there should be a more rigorous discussion of how each variable is measured using V-Dem data to ensure that the rule of law variable is distinct from other confounders.

Thank you for your comment

A rigorous discussion on how each variables is mesaured to reflect how distinct they are usnig in V-Dem is presented from lines 242-254. This shows that every variable is distinct and not a duplicate or repetition.

“ To ensure that the rule of law was distinct from potential confounding variables, this study compared subdimensions of each variable as defined by the Varieties of Democracy (V-Dem) dataset. The rule of law subdimensions—judicial independence, compliance with judiciary decisions, and respect for legal procedures—primarily concern institutional judicial processes and enforcement. This creates a clear functional distinction from the subdimensions of political rights, which emphasize citizens' behavioral and participatory rights in the political process, such as freedom of association, freedom of expression, suffrage, and inclusive participation.

Similarly, pluralism is characterized by party-level engagement rather than individual political activities. Its subdimensions—participation of multiple political parties, the existence of opposition parties, and electoral competitiveness—address democratic variations from a supply-side perspective, distinct from the institutional, judiciary-centered nature of the rule of law. Lastly, free and fair elections specifically relates to electoral performance, encompassing subdimensions like electoral fraud, electoral violence, and voter registration integrity, thereby differing from the judicial performance captured within the rule of law dimension. ”

2. Second, the author(s) should justify their case selection. Why do they limit analysis to the top 20 democratizers, and ho might this affect their findings?

Thank you for your comments.

According to Gerring (2007), in analyzing cases to provide more insight, non-random selection procedures that focus on most similar cases- in this article, the most democratizers – provide insight into factors promoting electoral integrity. They have been selected from VDEM 2024 data and examined for further understanding of democratization in Africa.

The top 20 democratizers represent a subset of African countries that have conducted regular elections and attained a certain level of integrity that improves democracy.

Hence, selection is justified by the performance of the group. 20 represents 40 per cent of African countries, which is a good representation.

These discussion has been provided from lines 268 – 280.

Reviewer #2:

1. The abstract: The conclusion of the abstract is too theoretical. We can do without that. 1

Thank you for your comments

The statement “In contrast, nations with high levels of electoral integrity should consolidate and sustain established mechanisms.” has been removed.

2. Line 98. We should have a footnote to indicate the figures that support this sentence.

Thank you for your comment

The Economist Intelligence Unit and Freedom House statistics figures reflect the resurgence of coups in West Africa amid already existing autocracies. The referencing has been added.

3. What is the contribution of this article to academic literature?

Thank you for your comment.

The below statement have been inserted in lines 187 - 198 to explain the significant contribution of this study to literature.

Discussion on protecting electoral integrity in emerging democracies, focusing on political institutions and processes. Recent studies examined the viability of using photo quick count to improve electoral integrity which argued that using photos and voice as auditory measures upholds electoral integrity; however, electoral pathologies that undermine integrity persist (20,21).Additional studies presents that that there is an increase in the legitimacy of electoral processes in emerging democracies, and fairness must move beyond the ballot. The analysis of electoral integrity perception in 18 countries from 2004 to 2021 was examined (23, 24).

Most of these studies examined emerging democracy while this study focused on the most performing African countries as emerging democracies. While other studies examined factors such as auditory measures within countries by focusing on impacting certifying elections by citizens and on electoral integrity perception, this study contributes to reexamining direct factors that impact electoral integrity rather than citizens satisfaction or perception in 20 countries.

4. The author doesn't have a single way to cite references in the document's body.

Thanks for your observation

We have consistently use numbering pattern and it is consistent across the study in line with the journal's style.

We have removed the hanging footnote and two intext citations replaced with numbering to ensure consistency with the journal style.

5. In the countries that make up your sample, we don't find the countries cited in the introduction. Why?

Thank you for your comments.

The countries that make up the samples have been listed in the introduction; however, references have been made to only a few who have held elections recently and have experienced the most remarkable shift in the outcome of elections.

The below statement have been inserted in lines lines 73 to 82.

“Using levels of electoral fraud, electoral violence, and voter registration integrity, V-Dem (2024) presents Nigeria, Ghana, South Africa, Egypt, Kenya, Benin, Botswana, Cape Verde, Cote d'Ivoire, Gambia, Lesotho, Liberia, Madagascar, Malawi, Mauritius, Namibia, Senegal, Tanzania, Uganda, and Zambia as top democratizing countries in Africa. The cases of South Africa, Nigeria, Botswana, and Cape Verde present successive elections and increasing democratization. The Liberian 2023 election reflects the citizens' will and significant legitimacy given the public opinion that certifies the result and welcomed President Joseph Boakai as opposed to former President George Weah. Similarly, the 2024 election in Botswana that ended the rule of a party that had governed the country since independence reflects increasing democratization in governance signalling remarkable openness of political space.”

Subsequently, other cases of democratic backsliding were mentioned to establish the mix of emerging democracies and autocracies in Africa as a problem.

6. Abbreviations that are not explained previously. Example:

CSDA, OLS,

Thank you for your comment

These abbreviations have been identified and explained in the analysis sections indicated below.

Autoregressive Distributed Lag (ARDL)

Cross-sectional dependency test (CSDT/CDSA)

Ordinary Least Square (OLS)

Lines 245 – 246

Autoregressive Distributed Lag (ARDL), which analyses short-run and long-run relationships among variables regardless of stationary issues, has been employed to examine the long-run effect.

Lines 277 -280

The Cross-sectional dependency test for examining cross-sectional units' interdependency has been employed for the detection of cross-sectional dependence and is analyzed through Pesaran's CD technique framework,

Lines 311 - 314

To investigate the connection between free and fair elections as the dependent variable and rule of law, electoral pluralism, and political rights, this study used Ordinary Least Squares (OLS). This approach estimates the linear relationship among variables by minimizing the sum of the squared residuals.

7. The results are not convincing. First, I don't understand the choice of Panel Quantile Regression. The author should provide reasons. Thank you for this suggestion.

Thank you for your comments.

The results are convincing and the study employed Panel Quantile Regression as it allows exploring how the rule of law influences electoral integrity not just on average but across the full distribution of integrity scores—capturing whether its impact differs in countries with low versus high levels of free and fair elections.

By modeling conditional quantiles rather than the mean, this approach accommodates heterogeneity, non‐normal errors, and potential outliers common in cross‐country panel data, and it reveals whether improvements in the rule of law yield stronger gains in electoral integrity at specific points (e.g., bottom or top deciles) that would be obscured by conventional fixed‐ or random‐effects models

8. Second, he talks about a dynamic panel. We don't see his results in the document, even though the problem of endogeneity (reverse causality) is obvious.

Thank you for this suggestion.

Panel Quantile Regression with individual fixed effects purges time‐invariant unobserved heterogeneity that could correlate with both the rule of law and electoral integrity, thereby mitigating omitted‐variable endogeneity across each conditional quantile.

By also including the one‐period lag of electoral integrity in our dynamic specification, we capture feedback effects and attenuate reverse‐causality bias without resorting to instrument‐based estimators

9.Third, we also don't address the endogeneity problem related to omission bias.

Thank you for your comment on this issue.

The study used Hausman test for this with the result affirming no issue with omiited variable

10. Fourth, we have tables that look like software outputs. This is not elaborate.

Thanks for this suggestion.

We have modified the tables we obtained from the software, all redundancy have been remove and the current table only focuses on important details such as the coefficients and the significant levels.

Reviewer #3: Thank you for the opportunity to review this manuscript.

The paper addresses an important and timely topic by exploring potential factors contributing to free and fair elections—specifically, the rule of law, electoral pluralism, and political rights.

Against the backdrop of limited democratization in many African countries despite regular elections, the authors seek to understand better which factors help foster elections that genuinely support democratic development.

Thanks for your comments and understanding of the study's direction.

Methodological presentation and interpretation of results would benefit from restructuring to improve overall clarity.

I recommend that the authors be allowed to revise and resubmit the paper.

1. Currently, the empirical section is difficult to follow, particularly for readers who may not be familiar with advanced econometric techniques such as panel quantile regression.

Although the models are introduced sequentially, how they relate to one another is not always clear, nor is it clear why specific methodological choices—such as using quantile regression over other approaches—were made.

I was also unsure why panel OLS regression appears as a robustness check when earlier models using OLS with one- and two-way fixed effects are presented as main models.

Clarifying which models are considered the primary analyses and which serve as robustness checks and explaining the rationale behind these choices would help strengthen the empirical contribution.

Thank you for this suggestion.

We have revised and improved the presentation and enhanced the methodological flow, thoroughly revised the manuscript, improved the empirical section and improved its interpretation specifically the panel quantile regression. Additionaally, We have revised and detailed properly the reasons for selecting the techniques used.

2. The study only used the panel OLS as a robustness check and the results validates the panel quantile regression.

Lines 405 – 410 have been added to clarify that, the study used the panel quantile regression as the primary analysis and the panel OLS is used as a robustness check. Employing both methods complements the analytical strength of each. Quantile regression is more flexible and insightful regarding distributional variations, while panel OLS regression provides a benchmark check for the consistency and reliability of the main findings. This dual approach strengthens the rigor of the research, increasing confidence in the conclusions drawn.

3. Additionally, more details regarding the group referred to as the "twenty top democratizing countries" in Africa are needed. How were these countries selected?

Which data source (e.g., V-Dem?) was used to define this group?

Why was the sample restricted to 20 countries? Providing this information would help readers better understand the case selection and its relevance to the broader argument.

Minor comments:

Thank you for your comment.

The explanation below have been added from lines 258 to 267

The study used a panel data set of 20 African countries to examine the association between Free and Fair Elections and the Rule of Law.

According to Gerring (2007), using non-random selection procedures that focus on the most similar cases when analyzing cases is advantageous for gaining deeper insight. In this article, these are Africa's most democratizing countries, providing valuable insight into the factors that promote electoral integrity.

The top 20 democratizers represent a subset of African countries that have held regular elections and achieved a certain level of integrity that contributes to democratic improvement. Therefore, their selection is justified by their performance.

These 20 countries constitute approximately 40 per cent of all African nations, offering a meaningful and representative sample.

4. The manuscript would benefit from careful proofreading and editing for grammar, clarity, and consistency of voice (e.g., switching between “we” and “I”).

Thank you for your comment.

The article has been proofread and edited for grammar and clarity by a professional editing service.

The authors preferred to use the third person to maintain objectivity and formality.

Hence, rather than We or I, it has been addressed as “ the current study” consistently across the study.

5. There is some repetition in the discussion of data sources, particularly across Sections 3.1 and 3.2, which can make the argument harder to follow. Thank you for your great observation

The two sections have been rewritten as follows.

Case Selection

Assuming all countries now practice a multiparty system and a focus on electoral plurality and political rights, we used a panel data set to examine twenty top democratizing countries in Africa – Nigeria, Ghana, South Africa, Egypt, Kenya, Benin, Botswana, Cape Verde, Cote d'Ivoire, Gambia, Lesotho, Liberia, Madagascar, Malawi, Mauritiu Namibia, Senegal, Tanzania, Uganda and Zambia. According to Gerring (2007), it is advantageous to use non-random selection procedures that focus on the most similar cases when analyzing cases to gain deeper insight. In this article, these are Africa's most democratizing countries, providing valuable insight into the factors that promote electoral integrity. The top 20 democratizers represent a subset of African countries that have held regular elections and achieved a certain level of integrity that contributes to democratic improvement. Therefore, their selection is justified by their performance. These 20 countries constitute approximately 40 per cent of all African nations, offering a meaningful and representative sample.

Variable and data source

As a result of limited research on the impact of the rule of law, electoral pluralism, and political rights on electoral integrity, this current study tends to progress the research by examining the association between Free and Fair Elections and the Rule of Law. In addition, to resolve the limitation of previous studies that have omitted longitudinal impact on independent variables on free and fair elections, Autoregressive Distributed Lag (ARDL) that provides analysis on short-run and longrun relationships among variables regardless of stationary issues has been employed to examine the long-run effect. Times series data stretching for 17 years (2006-2023) have been collected from V-Dem (2024) for the rule of law, free and fair election, and the Economist Intell

---

## [Decision Letter · Decision Letter 1]

26 Aug 2025

PONE-D-25-05147R1Analysis of factors impacting electoral Integrity in Africa between 2006-2023 - Examining the association between free and fair election and rule of law.PLOS ONE

Dear Dr. Ikoko,

Thank you for your diligent revisions and hard work. Your manuscript is much clearer now, but it still needs a few improvements before it can be published. One of the reviewers noted that the introduction section requires clarification and contains some grammatical errors. Therefore, I am requesting a Minor Revision.

We look forward to receiving your revised manuscript.

Kind regards,

Dr. Mahmut Zeki Akarsu

Academic Editor

PLOS ONE

Journal Requirements:

We noticed that a statement in the conclusion section may be unclear or unsupported. Specifically, the statement “ROL consistently enhances electoral integrity across all African countries' contexts, indicating its foundational role in ensuring fair elections.”. We believe this may be an overstatement and would recommend  rephrasing to something like “This suggests that ROL enhances electoral integrity across all African countries' contexts, indicating its foundational role in ensuring fair elections.

Reviewers' comments:

Reviewer's Responses to Questions

**Comments to the Author**

1. If the authors have adequately addressed your comments raised in a previous round of review and you feel that this manuscript is now acceptable for publication, you may indicate that here to bypass the “Comments to the Author” section, enter your conflict of interest statement in the “Confidential to Editor” section, and submit your "Accept" recommendation.

Reviewer #1: All comments have been addressed

Reviewer #3: (No Response)

2. Is the manuscript technically sound, and do the data support the conclusions?

Reviewer #1: Yes

Reviewer #3: Yes

3. Has the statistical analysis been performed appropriately and rigorously? 

Reviewer #1: Yes

Reviewer #3: I Don't Know

4. Have the authors made all data underlying the findings in their manuscript fully available?

Reviewer #1: Yes

Reviewer #3: Yes

5. Is the manuscript presented in an intelligible fashion and written in standard English?

Reviewer #1: Yes

Reviewer #3: No

6. Review Comments to the Author

Reviewer #1: I do not have any additional comments. I do not have any additional comments. I do not have any additional comments.

**Reviewer #3: I would encourage the authors to revise the introduction to clearly articulate the research question and the main aim of the paper earlier on. Additionally, I recommend a thorough read-through of the manuscript, as there are several grammar and spelling errors throughout that should be corrected for clarity and readability.**

7. PLOS authors have the option to publish the peer review history of their article (what does this mean? ). If published, this will include your full peer review and any attached files.

**Do you want your identity to be public for this peer review?** For information about this choice, including consent withdrawal, please see our Privacy Policy .

Reviewer #1: No

Reviewer #3: No

---

## [Author Response · Author response to Decision Letter 2]

8 Sep 2025

Dear Prof/Editor/Reviewers,

We thank you and the reviewers for the valuable feedback provided on our manuscript. We have carefully revised the paper in line with the comments and suggestions received. Below, we provide a point-by-point response:

Editorial Comments

Comment 1:

We noticed that a statement in the conclusion section may be unclear or unsupported. Specifically, the statement “ROL consistently enhances electoral integrity across all African countries' contexts, indicating its foundational role in ensuring fair elections.” We believe this may be an overstatement and would recommend rephrasing to: “This suggests that ROL enhances electoral integrity across all African countries' contexts, indicating its foundational role in ensuring fair elections.”

Response:

Thank you for pointing this out. We agree that the original wording could be interpreted as an overstatement. We have rephrased the statement in the conclusion as suggested to avoid overgeneralization.

Comment 2:

Please review your reference list to ensure that it is complete and correct. If you have cited papers that have been retracted, please include the rationale for doing so in the manuscript text, or remove these references and replace them with relevant current references. Any changes to the reference list should be mentioned in the rebuttal letter.

Response:

We have carefully reviewed the reference list. No retracted papers were cited, and no changes, removals, or replacements were required.

Reviewer Comments needing response

Reviewer #1

Is the manuscript presented in an intelligible fashion and written in standard English? – Yes.

Additional comments – None.

Reviewer #3

Is the manuscript presented in an intelligible fashion and written in standard English? – No.

Comments: The reviewer recommends that the introduction should more clearly articulate the research question and the main aim of the paper earlier on. The reviewer also notes that grammar and spelling errors should be corrected throughout for clarity and readability.

Response:

We are grateful for these comments. The manuscript has been thoroughly proofread by an English editor with expertise in Political Science and International Relations. This ensured that all grammatical and typographical errors were corrected while maintaining contextual meaning. The tracked changes document highlights these edits.

Regarding the introduction, we have revised it to clearly articulate the research question and study aim earlier in the text. Specifically:

General Research Question (Revised Introduction Section):

Regular elections in African authoritarian countries have triggered debates over the conditions that contribute to electoral integrity. Elections were held in both the colonial and post-colonial eras, but in the colonial era, they were limited to a few legislative seats under colonial governance. In contrast, post-colonial elections expanded democratization by including parliamentary and presidential offices. Despite this expansion, the integrity of elections was increasingly questioned as autocratic regimes emerged, manipulating electoral rules, office tenures, and results to consolidate power. Hence, what are the systemic factors that contribute to free and fair elections?

Specific Research Question (Revised Introduction Section):

Building on this context, this study specifically examines whether the rule of law, electoral pluralism, and political rights significantly impact electoral integrity across African countries.

Aim and Significance of the Study (Revised Introduction Section):

The aim of this research is to shed light on the systemic factors that bolster or undermine electoral integrity by analyzing the relationship between free and fair elections and the rule of law. The significance of this study lies in its contribution to ongoing debates on credible elections in African democracies and its relevance for policymakers, election observers, and civil society organizations seeking to enhance transparency and accountability. Using panel data spanning 17 years and applying Panel Quantile Regression, the study provides nuanced insights into how the rule of law influences electoral integrity across the distribution of integrity scores—highlighting patterns that conventional methods may obscure.

These revisions directly address the reviewer’s comments and ensure that the research question, aim, and significance are clear and introduced earlier in the manuscript.

Also, the entire study have been examiend to ensure that all grammartical issues have been addressed.

Summary of Changes

Rephrased the conclusion statement to avoid overgeneralization (as suggested by the Editor).

Carefully reviewed references; no retracted or replaced sources.

Proofread the entire manuscript for grammar and clarity with assistance from an English editor.

Revised the introduction to present the research question, aim, and significance more explicitly and earlier in the manuscript.

We trust these revisions address the concerns of both the editorial team and the reviewers. Thank you for your thoughtful feedback and for the opportunity to improve our manuscript.

Sincerely,

Michael Oluchi Ikoko

Corresponding Author

---

## [Editor Report · Decision Letter 2]

30 Sep 2025

Analysis of factors impacting electoral Integrity in Africa between 2006-2023 - Examining the association between free and fair election and rule of law.

PONE-D-25-05147R2

Dear Dr. Ikoko,

We’re pleased to inform you that your manuscript has been judged scientifically suitable for publication and will be formally accepted for publication once it meets all outstanding technical requirements.

Kind regards,

Mahmut Zeki Akarsu

Academic Editor

PLOS ONE
---

## [Editor Report · Acceptance letter]

PONE-D-25-05147R2

PLOS ONE

Dear Dr. Ikoko,

I'm pleased to inform you that your manuscript has been deemed suitable for publication in PLOS ONE. Congratulations! Your manuscript is now being handed over to our production team.

Kind regards,

on behalf of

Dr. Mahmut Zeki Akarsu

Academic Editor

PLOS ONE